# Impact of Nutrient Intake on Body Composition in Very Low-Birth Weight Infants Following Early Progressive Enteral Feeding

**DOI:** 10.3390/nu16101462

**Published:** 2024-05-13

**Authors:** Rasa Brinkis, Kerstin Albertsson-Wikland, Kastytis Šmigelskas, Aurika Vanckavičienė, Ilona Aldakauskienė, Rasa Tamelienė, Rasa Verkauskienė

**Affiliations:** 1Department of Neonatology, Lithuanian University of Health Sciences, 44307 Kaunas, Lithuania; ilona.aldakauskiene@lsmuni.lt (I.A.); rasa.tameliene@lsmuni.lt (R.T.); 2Department of Physiology/Endocrinology, Institute of Neuroscience and Physiology, Sahlgrenska Academy, University of Gothenburg, 40530 Gothenburg, Sweden; kerstin.albertsson.wikland@gu.se; 3Health Research Institute, Faculty of Public Health, Lithuanian University of Health Sciences, 44307 Kaunas, Lithuania; kastytis.smigelskas@lsmuni.lt; 4Department of Nursing, Lithuanian University of Health Sciences, 44307 Kaunas, Lithuania; aurika.vanckaviciene@lsmuni.lt; 5Institute of Endocrinology, Lithuanian University of Health Sciences, 44307 Kaunas, Lithuania; rasa.verkauskiene@lsmuni.lt

**Keywords:** preterm, very low birth weight, newborn, nutrient intake, body composition, dual X-ray absorptiometry

## Abstract

Preterm infants have increased body adiposity at term-equivalent age and risk of adverse metabolic outcomes. The aim of the study was to define how nutrient intake may impact body composition (BC) of very low-birth weight infants fed with early progressive enteral feeding and standard fortification. Eighty-six infants with <1500 g birth weight were included in the BC study and stratified into extremely preterm (EP) and very preterm (VP) groups. Nutrient intake was calculated during the first 28 days and BC assessed by dual X-ray absorptiometry at discharge and by skinfold thickness at 12 months of corrected age (CA). Total nutrient intake did not differ between the groups. EP infants had a higher fat mass percentage at discharge than VP infants (24.8% vs. 19.4%, *p* < 0.001); lean mass did not differ. None of the nutrients had any impact on BC of EP infants. Protein intake did not result in a higher lean mass in either group; fat intake was a significant predictor of increased fat mass percentage in VP infants at discharge (*p* = 0.007) and body adiposity at 12 months of CA (*p* = 0.021). Nutritional needs may depend on gestational age and routine fortification should be used with caution in more mature infants.

## 1. Introduction

Numerous studies have revealed an altered body composition of preterm infants at term equivalent age compared to their term counterparts [1] and increased risk of metabolic consequences later in life [2,3,4]. Current feeding practices aim at a preterm body composition similar to that of term infants. However, it is challenging, if not impossible, to reach this target due to the extrauterine environment being much different from the intrauterine environment. For intrauterine growth, the foetus receives nutrients and oxygen delivered via the placenta and body fat content increases with gestational age [5]. Body fat storages become crucial for extrauterine survival of infants [6], and the fat mass percentage increases even more over the first months of life [7,8]. Thus, fat mass accumulation in early life is physiological and may be inevitable for preterm infants growing ex utero and being fed with human milk. However, excess fat accumulation in early life may result in metabolic consequences later in life [9,10], and lean mass accretion is associated with better neurocognitive outcomes [11,12]. Many factors may influence further developmental and metabolic disturbances in this population, with nutrition being the most modifiable factor. There is no doubt that human milk (HM) is the best food for all infants; however, for those born prematurely, human milk alone may not ensure adequate nutrient intake and requires fortification. It is known that for term breastfed infants, breast milk composition may influence body composition, with higher fat and energy content resulting in higher fat mass gain; however, healthy infants may regulate the breastmilk and nutrient intake through satiety mechanisms [13]. These mechanisms are absent in gavage-fed very preterm infants; moreover, nutritional human milk composition is altered by fortifiers. To date, there is no evidence available that macronutrient supplementation of HM improves outcomes of preterm infants [14,15,16]. Differences in HM and multi-nutrient fortifier composition results in variability of nutrient proportions in the fortified milk. With the increasing awareness of childhood obesity, the goal is to balance between improving neurodevelopment and preventing cardiometabolic consequences later in life. Recently, the understanding of preterm newborn physiology, growth, and nutritional needs has been changing, replacing the “foetus-like” growth concept with the establishment of postnatal, preterm growth standards [17,18,19], and leading to attempts at early feeding with human milk.

The aim of our study was to define the impact of early nutrient intake (during the first 28 days of life) on body composition of very low-birth weight infants fed with early progressive enteral feeding with human milk and standard human milk fortification.

## 2. Materials and Methods

### 2.1. Study Subjects

A non-interventional observational study was conducted at the Hospital of Lithuanian University of Health Sciences, Department of Neonatology in 2018–2022. One hundred and twenty infants with a birth weight of <1500 g and gestational age of ≤34 weeks without chromosomal abnormalities and major birth defects were included into the study. Approval for the study was obtained at the Kaunas Regional Bioethics Committee (approval no. BE-2-12) and the study was registered in the ISRCTN Database (No. ISRCTN64647571). Both parents signed informed consent forms. During the hospital stay, 8 infants died, and 3 infants were excluded from further follow-up due to necrotizing enterocolitis and subsequent partial bowel removal. At discharge, which occurred at the median age of 36 weeks of corrected gestational age (CGA), a whole-body dual X-ray absorptiometry (DXA) scan was performed on 93 infants. After exclusion of scans which were considered inaccurate due to excessive movement artefacts, 86 scans were included into the final analysis. Scanning was impossible with certain medical conditions, such as supplemental oxygen or an orthopaedic cast at discharge. A total of 11 infants missed the scan due to early COVID-19 restrictions for non-emergency procedures and DXA machine technical problems. 

For a more detailed analysis, the cohort was stratified into two gestational age groups: extremely preterm infants, EP (born at 23–27 gestational weeks), and very/moderately preterm infants, VP (born at 28–33 gestational weeks). The flow chart of the study subjects and reasons of exclusion from DXA analysis are presented in Figure 1.

### 2.2. Feeding Practices and Nutritional Calculations

All infants were started on both parenteral and enteral nutrition right after birth. Enteral nutrition was started at 20 mL/kg/day with oral application of own mother’s colostrum and advanced rapidly by 20–30 mL/kg/day with fresh own mother’s milk and donor milk if mothers’ lactation was not fully established. Feeding tolerance and advancement rate was defined by the attending physician. Parenteral nutrition was gradually weaned with the increasing enteral milk volume. After reaching full enteral feeding, standard human milk (HM) fortified with multi-nutrient bovine-milk based fortifier (Aptamil FMS^®^, Milupa/Danone GmbH, Friedrichsdorf, Germany, Danone Nutricia, Cuijk, The Netherlands) was started according to the manufacturer’s recommendations, i.e., 4.4 g of fortifier was added to 100 mL of milk. The fortification was continued until discharge and no post-discharge feeding interventions were administered. HM composition was measured twice a week using a mid-infrared spectroscopy HM analyser (MIRIS, Uppsala, Sweden) and actual daily nutrient intake was calculated. Detailed nutritional practices and nutrient intake calculations have been described previously [20]. Mother’s milk was analysed only after there was enough of it to feed the infant. The benefits of colostrum and early own mother’s milk were considered superior over nutritional analysis. Total nutrient intake (enteral and parenteral) was calculated during the full first 4 weeks and used for regression analysis. Only known values of nutrients in HM were used in calculations; therefore, only a few first week enteral intakes from mother’s milk could be included and average nutrient intake for the whole 28-day period was slightly underestimated.

### 2.3. Body Composition Assessment and Anthropometric Measurements

Body composition was measured by a dual energy X-ray absorptiometry (DXA) scanner, Hologic Discovery, model Horizon A (Hologic Inc., Marlborough, MA, USA) using infant whole body software (Version APEX v.5.6.0.5). The median age at the DXA scan was 36 weeks of postmenstrual age (PMA). The infant was scanned lying supine during natural sleep, usually after feeding. All infants were dressed in the same type of light clothing and disposable diaper and swaddled into the same type of light blanket. Each infant was scanned only once, and scans with excessive movement artefacts were excluded from final analysis. Total body mass, lean mass, fat mass, and fat mass percentage were evaluated. Lean mass values were estimated as lean mass plus bone mineral content (LM + BMC). There was no possibility of performing DXA analysis without sedation at 12 months of CA; thus, skinfold thickness measurements at four body sites were used to assess body adiposity at the follow-up day and the day of the DXA scan.

Skinfold thickness (SFT) measurements were performed using a Harpenden calliper (Baty International, Burgess Hill, UK). The Harpenden calliper operated with a pressure of 10 g/mm^2^, to the nearest measure of 0.1 mm. Measurements were duplicated and the average value was used for analysis. SFT was measured at 4 sites on the right side of the body: biceps, triceps, subscapular, and suprailiac. Biceps SFT was measured in the middle of the upper arm, parallel to the long axis with the arm extended; triceps SFT was measured in the middle of the upper arm, parallel to the long axis with the arm flexed. Subscapular SFT was measured right below the angle of right scapula and suprailiac SFT was measured above the iliac crest at the mid-axillary line. Pressure was applied until the reading was stable [21,22]. The mathematical sum of skinfold thickness was used to estimate body adiposity at the day of the DXA scan and at 12 months of CA. Skinfold thickness at 12 months of CA was measured in 30 EP and 45 VP infants, with the dropouts occurring due to COVID-19 restrictions.

Anthropometric measurements were performed following the standard procedure [23]. Infants were weighed using electronic infant scales (Marsden, Rotherham, UK). Length was measured with an infant measuring rod (SECA, Hamburg, Germany). 

### 2.4. Statistical Analysis

Statistical analyses were performed using Microsoft Excel version 16.81 and IBM SPSS Statistics for Windows (version 29.0.1.0, IBM Corp., Armonk, NY, USA). Descriptive analysis for normally distributed variables included means and standard deviations (SD); for those without normal distribution—medians and interquartile ranges (IQRs); for categorical indicators—percentages. The normality of variable distribution was estimated using skewness and kurtosis. For group comparisons, the parametric Student’s *t*-test for normally distributed values and the non-parametric Mann–Whitney U test for values without normal distribution were employed. For comparison of categorical indicators, the Chi-squared test was used, and the Fisher’s exact test was used when the conditions for the Chi-squared test were not met. The results of dependent measurements were compared using the paired-samples *t*-test. The relationships between two continuous variables were assessed using the Pearson correlation coefficient.

For prediction of total body mass, lean body mass, fat mass, fat mass proportion, and body adiposity at 12 months of CA expressed by sum of SFT, multivariable linear regression models were created. The effect size of the factors in the model was evaluated in terms of standardised beta coefficients (β_s_). Model fit was estimated through the determination coefficient R^2^. The statistical significance was set at *p* < 0.05.

## 3. Results

The main demographic characteristics of the infants included in the final analysis are described in Table 1.

### 3.1. Nutritional Characteristics

Enteral feeding was started within the first 6 h after birth for 80% EP and 98% VP infants. Median time to reach full enteral feeding was 7 days in the whole cohort. EP infants received longer parenteral support and had higher median parenteral protein, carbohydrate, and energy intake than VP infants, while the fat intake did not differ between the groups. EP infants were started on enteral feeding later and the fortification was introduced later compared to VP infants. Enteral intake during weeks 2 to 4 did not differ between the groups, and nor did total intake during the first four weeks. Total nutrient intake did not differ between boys and girls (*p* = 0.865 for protein, *p* = 0.927 for carbohydrates, *p* = 0.435 for fat, and *p* = 0.556 for energy intake). The main nutritional characteristics are presented in Table 2.

### 3.2. Body Composition Estimated by DXA at the Hospital Discharge

DXA-estimated mean body mass was higher than the infants’ actual body weight at the DXA scan day (2614 (±393) g vs. 2386 (±378) g in the whole cohort, t = 34.30, *p* < 0.001); however, the correlation between actual weight and DXA-estimated weight was very good (r = 0.99, *p* < 0.001). Boys had a significantly higher lean body mass at discharge than girls in the whole cohort (2164 (±247) g vs. 1945 (±208) g; t = 4.46, *p* < 0.001) but fat mass and fat mass percentage did not differ between the sexes (616 (±217) g vs. 543 (±238) g; t = 1.46, *p* = 0.147 and 21.7 (±5.5)% vs. 21.2 (±7.2)%; t = 0.34, *p* = 0.731, respectively). Time at DXA scan (PMA), weight on the DXA scan day, and DXA data of total body mass, lean mass, fat mass, and fat mass proportion of all subjects by gestational age are shown in Figure 2.

Extremely preterm infants were scanned at the median chronological age of 10.2 weeks and PMA of 36.2 weeks (Appendix A) and very preterm infants were scanned at the median chronological age of 6.2 weeks and PMA of 35.4 weeks. EP infants had a higher body weight at discharge than VP infants (2558 g vs. 2284 g, *p* < 0.001) and a higher DXA-estimated whole body mass (2876 g vs. 2512 g, *p* = 0.001). Lean mass was similar in both groups (2086 g in EP and 2012 g in VP, *p* = 0.188), while the fat mass and fat mass percentage (%) increased with gestational age and was significantly higher in EP infants than in VP infants (700 g and 24.8% vs. 500 g and 19.4%, *p* < 0.001). The sum of skinfold thickness measured at discharge highly correlated with DXA-estimated body fat percentage (r = 0.59, *p* < 0.001); thus, it was used as a body adiposity estimate at 12 months of CA. By 12 months of CA, there was no difference in body adiposity expressed by the sum of SFT between EP and VP infants (26.9 and 26.3 mm, *p* = 0.661).

The main differences in body composition values between the two groups are presented in the Appendix A.

### 3.3. The Association between Early Nutrient Intake and Body Composition

Multivariable regression was conducted to see how nutrient intake may impact the body composition at discharge and at 12 months of CA (Table 3). Models for DXA characteristics included total average nutrient intake during the first 28 days, sex, and PMA. The model for body composition at 12 months of CA included total average nutrient intake during the first 28 days, sex, and gestational age. Energy was not included into the models due to collinearity. In the group of EP infants, none of the macronutrients had any effect on total mass, lean mass, fat mass or fat mass percentage at discharge, and body adiposity expressed as the sum of SFT at 12 months of CA. Higher protein intake did not result in a higher lean body mass in either group at both assessment time points. Higher carbohydrate intake during the first 28 postnatal days did not affect body composition in either group at discharge and at 12 months of CA; however, a higher fat intake in VP infants was a significant predictor of a higher absolute fat mass (β_s_ 0.39, *p* = 0.007) and fat mass percentage (β_s_ 0.43, *p* = 0.005) at discharge, and body adiposity expressed as the sum of SFT at 12 months of CA (β_s_ 0.41, *p* = 0.021).

At discharge, male sex was associated with a higher lean mass in both groups (β_s_ −0.42, *p* = 0.021 in EP and β_s_ −0.30, *p* = 0.030 in VP infants). A higher PMA was associated with a higher total body mass and lean mass in both groups. Fat mass was positively associated with PMA in VP infants; however, fat mass proportion did not reach statistical significance. Regression analysis results are presented in Table 3.

## 4. Discussion

### 4.1. Body Composition of Preterm Infants at Discharge and at 12 Months of CA

Assessment of body composition in the preterm population is gaining more interest since it is a valuable tool to evaluate quality of growth together with anthropometric measurements. However, little knowledge is present about ideal body composition for preterm infants growing ex utero. Current body composition references for preterm infants are based on cross-sectional measurements of preterm infants soon after birth measured by air displacement plethysmography (ADP) [5]. ADP is one of the validated methods of body composition assessment in infants and is more commonly used in research than DXA [25]. A study by Yumani et al. compared the results of ADP, DXA, and clinical body adiposity assessment by skinfold measurements and found poor agreement between ADP and DXA, with DXA giving higher estimates of fat mass and fat mass percentage than ADP [26]. Blankets and ingested milk may influence DXA results in infants [27], and all our study patients were examined when swaddled and fed. This may explain the difference between actual weight and DXA-estimated total body mass; similar differences are reported in other DXA studies [28,29].

Infants in our cohort receiving early progressive enteral feeding with own mother’s milk had increased body adiposity compared to term reference values from published studies where fat mass percentage measured by DXA in term infants ranged from 11% to 13% [27,30]. One of the determinants of a higher body adiposity in EP infants in our study may be a longer exposure to the extrauterine environment, as they had a similar fat percentage at 10 weeks after birth (24.8%) compared to term infants at 2 months of age (24.5%) in the study by Schmelzle at al. [30]. VP infants in our study were scanned at the median age of 6 weeks, which may explain their lower fat mass percentage.

Our results are consistent with a similar prospective observational study conducted in Austria, where infants born at <32 gestational weeks received nutrition according to ESPGHAN 2010 guidelines and body composition was assessed using PEA POD air displacement plethysmography close to term-equivalent age [31]. Extremely preterm infants in this study had a higher fat mass than very preterm infants—17% versus 15.5%. Like in our cohort, very preterm infants were discharged and scanned earlier than extremely preterm infants, and VP infants had a lower weight at discharge. The Austrian study did not analyse associations between nutrients and body composition, and parenteral support was much longer compared to our cohort (28 days in extremely preterm and 9 days in very preterm infants). Another study in Germany included 105 preterm infants who were started on enteral feeding right after birth, just like in our cohort; feeding was advanced rapidly and time to reach full enteral feeding was 5 days, even shorter than in our study [32]. Body composition was assessed using ADP at 37.6 weeks. The cohort was stratified to EP and VP infants and SGA and AGA infant groups. EP infants were examined at a later postnatal and postmenstrual age than VP infants and, contrary to our findings, fat mass and fat mass percentage did not significantly differ between these groups. PMA at BC assessment in this study was a significant predictor of fat-free mass, fat mass, and fat mass percentage, consistent with our findings. Fat-free mass was similar in infants of the German cohort to our study infants, even though examined by different techniques. Interestingly, other previous studies found good agreement between ADP and DXA-estimated fat-free mass [26,33]. The above-mentioned study examined the influence of parenteral protein intake during the first week of life on body composition; however, no associations were found. We analysed average daily nutrient intake throughout the 28-day period and did not find any associations between protein intake and fat mass or fat-free mass at discharge in either group of infants. A randomised trial by Sallas et al. [34] reported that increased protein intake with additional fortification resulted in higher fat-free mass z-scores in extremely preterm infants at discharge; however, at 3 months of CA the fat-free mass difference between higher and lower protein groups was not statistically significant. Moreover, higher protein did not result in a lower fat mass percentage in that study. A higher protein/energy ratio was found to be significantly associated with a decreased risk of fat-free mass deficit in the study by Simon [35]; the protein/energy ratio was 2.5–2.6 g/100 kcal, while in our study it was 3.1 g/100 kcal in both groups. This may suggest that there is a certain threshold for protein/energy intake, above which lean mass accretion will not be improved. Another finding in that study was that beside nutrition, male sex was associated with a higher risk of fat-free mass deficit, whereas in our study male sex was a significant predictor of a higher absolute lean body mass in both groups. Boys in our cohort also had a higher lean mass close to the term-equivalent age even though nutrient intake was not different between boys and girls. Fat mass and fat mass percentage did not differ between sexes, and this is consistent with the finding of another study by Simon et al., which found that fat mass percentage did not differ between preterm boys and girls, while term male infants had a higher fat-free mass and lower fat mass percentage than females [36].

In our cohort, despite a significant difference in fat mass at discharge between two gestational age groups, this difference was not maintained by 12 months of CA. It is unclear whether this happened by the slowing down of fat mass accumulation in the extremely preterm infants or the opposite process in more mature infants. These results should be interpreted with caution, since at follow-up we could apply only clinical methods of adiposity evaluation, i.e., skinfold measurements.

### 4.2. The Association between Early Nutrient Intake and Body Composition

Our study reveals differences in the impact of early nutrition on the body composition in infants born at different gestational ages. With early enteral feeding with own mother’s milk, nutritional deficits may be minimised, especially in infants born at higher gestational ages. With the variability of human milk composition, standard fortification, which is mostly used in the clinical setting, may provide too little or too much nutrients. The latest ESPGHAN recommendations on enteral nutrient intakes involve recommendations for all infants born <1800 g [37]. Extrauterine growth and development might be completely different in infants born at the earliest gestation and those who are more mature. Moreover, nutritional management is different for acutely sick infants and those recovering from acute illness or ones without major morbidities [38], and this practice was not always applied in infants of our study. In our cohort, VP infants had fewer morbidities compared to those born extremely premature, and growth patterns were different during the early postnatal period despite the average nutrient intake not being significantly different, which is described in a previous publication [39].

Duration of parenteral nutrition was short in VP infants—median 5 days; therefore, enteral fat intake was dominant during the first 28 postnatal days. It is important to note that the only source of enteral fat intake was human milk. Fortifier consisted of additional protein, carbohydrates, and micronutrients; however, it provided more carbohydrates than the recommended intake, especially in the group of VP infants. At the time of the study, recommended carbohydrate intake was 11.6–13.2 g/kg/day [40], while in the current recommendations the carbohydrate intake limit is increased up to 15 g/kg/day [37]. It is known that nutrient proportions in the diet of low-birth weight infants may influence the quality of growth. A randomised trial by Kashyap et al. [41] analysed different energy, carbohydrate, and fat proportions in infant formulas and reported better weight gain and nitrogen retention in the group of infants fed with the formula in which carbohydrates were dominant over fat as the non-protein energy source, suggesting that carbohydrates could be easier utilised for protein accretion than fat. Although this study was performed with infant formulas, the same principles might apply to feeding with fortified human milk. With extra carbohydrates in the fortifier in our study, we may hypothesise that some non-protein energy from fat intake may have been stored as excess body fat.

Associations between fat intake and body adiposity were reported in the earlier studies. In the study performed by Tremblay [28], authors found that intravenous lipids during the first week of life were positively associated with DXA-estimated abdominal fat mass at discharge; however, duration of parenteral nutrition in this study was longer (29 days), full enteral feeding was achieved later than in our cohort, and enteral intake was not assessed. Whether parenteral intake has the same effect on body composition as enteral intake is not known. Another study conducted by McLeod et al. [42] evaluated enteral nutrient intake effects on body composition in infants born <33 gestational weeks. Body composition was assessed using ADP. Mean enteral intake in subjects of this study was similar to ours, with breastmilk being the predominant nutrient source. The study revealed that fat intake was associated with increased absolute fat mass in grams, but not in percentage of body fat, while in our cohort associations were found in both absolute fat mass and percentage of body fat in the group of infants born at 28–33 weeks. Contrary to McLeod’s study, we did not find a positive association between protein intake and fat-free mass in either group. A higher lipid intake was associated with a higher percentage of body fat in a prospective cohort study of very preterm infants conducted by Han et al. [43], with actual nutrient intake estimation by human milk analysis. The mean time to achieve full enteral feeding was 21.5 days, much later than in our cohort, suggesting that solely enteral intake was not dominant. Important finding of this study is that the impact of higher lipid intake on fat mass persisted to 6 months of CA, which is consistent with our findings—at 12 months of CA in our VP infants, higher lipid intake predicted higher body adiposity measured by SFT. The impact of nutrient intake and nutrient proportions on body composition was also reported in the study by Lingwood [44], where it was found that higher protein and carbohydrate intake, and a higher protein/carbohydrates ratio were associated with a lower fat mass in infants born less than 32 weeks; nutrient intake was calculated using the reference values for human milk and more infants received unfortified human milk than fortified, i.e., the nutritional intervention was different from our cohort, which might explain the different results in our study.

The importance of human milk composition variations on fortification strategies and infant growth and body composition are well described in the studies by Belfort [45] and Rochow [46]. In the Belfort study, analysis of unfortified human milk showed remarkable variations between mothers and subsequent day-to-day variation in individual infants. Analysing the associations between unfortified human milk nutrient intakes and growth and body composition of very preterm infants, the authors found that more fat and energy from human milk was associated with a higher fat-free mass z-score. When compared to our study results, where fat intake was associated with more fat mass, it might be suggested that nutrients from unfortified and fortified human milk may have different effects on growth and body composition. In the randomised trial by Rochow and colleagues, human milk was analysed three times per week and the intervention group received standard fortification with additional targeted fortification to achieve recommended nutrient intakes, while the control group received standard fortification only. In this study, the fortifier used for standard fortification had a different macronutrient composition than that in our study; therefore, the nutrient intake proportions in the standard fortification group were not the same; the standard group received fewer carbohydrates and more fat than infants in our cohort: even with additional fortification in the intervention group, carbohydrate intake was lower than in our cohort. The intervention group had better growth, a higher fat-free mass, and a higher fat mass measured by ADP. Fat mass percentage did not differ between the groups, and it is not clear which nutrient resulted in more fat or fat-free mass. Authors of this trial discussed the importance of the natural variations in composition of the human milk and possible nutritional deficits as well as possible nutrient excess with the standard fortification.

In comparison with aforementioned studies, our study results may support the hypothesis that both human milk and fortifier composition may influence quality of growth in certain subgroups of very low birth weight infants. More mature and relatively “healthy” preterm infants, especially those whose mother’s milk has higher concentrations of nutrients, may not benefit from standard routine multinutrient fortification, and may face a risk of increased body adiposity with this effect observed even later in life. Longer term follow-up studies show that a nutrient-enriched diet and accelerated early growth may result in a higher body fat mass in childhood and are associated with increased adiposity and determinants of cardiovascular disease in young adults [47,48].

Our study has several limitations. The main limitation is the observational nature of the study and the relatively small single-centre cohort. Early COVID-19 restrictions and DXA machine maintenance issues resulted in a dropout of 11 infants at discharge; 11 infants did not return to the 12-month follow-up, making the final cohort even smaller. There were more girls than boys in the VP group (62% vs. 38%), which may have influenced the results; however, the small sample size did not allow us to stratify infants by sex. At 12 months of CA follow-up, we could not perform DXA scans without sedation; it was not possible to arrange the scan during natural sleep for older infants, and skinfold thickness might have been an estimate with low accuracy. Despite DXA being validated for infants, ADP is more widely used due to the absence of ionising radiation and ease of measurement [25]. For that reason, results of our study are difficult to compare with recent nutritional studies with similar feeding practices. In addition, we did not assess or control post-discharge feeding, which may have played a role in body composition at follow-up.

## 5. Conclusions

The results of our study suggest that preterm infants born at different gestational ages may have different nutritional needs; moreover, fortification without HM analysis may result in undernutrition as well as overnutrition of certain ingredients. Own mother’s milk should be the primary food for preterm infants, with careful choice of fortifiers—probably single-nutrient over multi-nutrient, depending on HM composition. In addition, the body composition of preterm infants is associated with postmenstrual age and time spent in the extrauterine environment from birth to term-equivalent age; thus, aiming for a foetus-like body composition with a relatively low fat mass at term-equivalent age may be an unachievable goal. Further studies should aim at finding postnatal growth patterns of preterm infants, ensuring the most favourable long-term outcomes.

## Figures and Tables

**Figure 1 nutrients-16-01462-f001:**
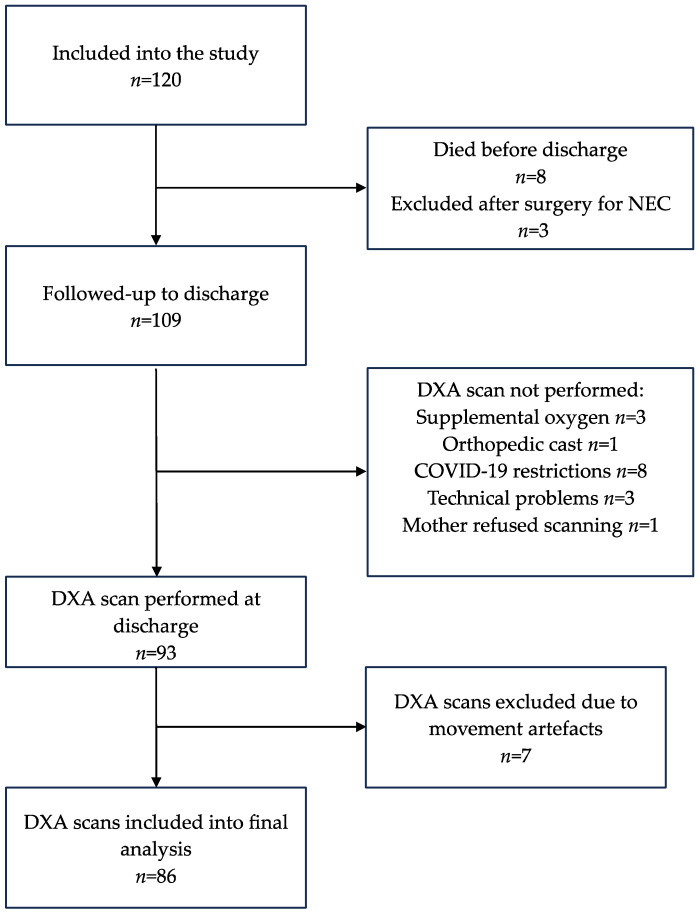
Flowchart of the study. DXA—dual X-ray absorptiometry, NEC—necrotising enterocolitis.

**Figure 2 nutrients-16-01462-f002:**
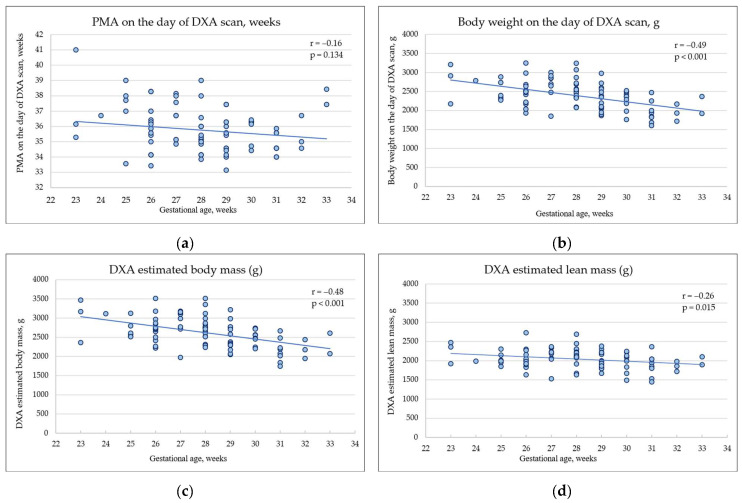
Postmenstrual age (PMA) on the day of DXA scan, body weight, DXA-estimated total body mass, lean mass, fat mass in grams, and fat mass proportion in percentage of all subjects. (**a**) PMA on the day of DXA scan, weeks; (**b**) body weight on the day of DXA scan, grams; (**c**) DXA-estimated total body mass, grams; (**d**) DXA-estimated lean mass, grams; (**e**) DXA-estimated fat mass, grams; (**f**) DXA estimated fat mass proportion, percent.

**Table 1 nutrients-16-01462-t001:** Main demographic characteristics of the infants included into final analysis.

Characteristics	Extremely Preterm *n* = 32	Very Preterm *n* = 54	*p*
Male sex, *n* (%)	16 (50.0)	21 (38.9)	0.314
Gestational age, weeks	26 (25–27)	29 (28–30)	<0.001
Birthweight g, mean (SD)	878 (±197)	1267 (±140)	<0.001
Apgar score 1 min	6 (3–8)	8 (7–8)	<0.001
Apgar score 5 min	8 (6–8)	8 (8–9)	<0.001
Caesarean section, *n* (%)	14 (43.8)	32 (59.3)	0.163
Multiple pregnancy, *n* (%)	5 (15.6)	18 (33.3)	0.073
SGA, *n* (%)	6 (18.8)	12 (22.2)	0.702
Mechanical ventilation, *n* (%)	16 (50.0)	2 (3.8)	<0.001
Sepsis *, *n* (%)	18 (56.3)	7 (13.0)	<0.001
PDA (HS), *n* (%)	18 (56.3)	8 (14.8)	<0.001
BPD, *n* (%)	3 (9.4)	1 (1.9)	0.143
NEC, *n* (%)	0 (0.0)	1 (1.9)	N/A

SGA—small for gestational age (Lithuanian national reference, 2023 [24]), PDA (HS)—hemodynamically significant patent ductus arteriosus, BPD—bronchopulmonary dysplasia, NEC—necrotising enterocolitis. * Sepsis—both early- and late-onset.

**Table 2 nutrients-16-01462-t002:** Enteral, parenteral, and total daily average nutrient intake in two gestational age groups.

Characteristics	Extremely Preterm *n* = 32	Very Preterm *n* = 54	*p*
Enteral feeding started, hours	5 (3–6)	3 (3–5)	0.005
Parenteral nutrition, days	7 (6–9)	5 (4–6)	<0.001
Fortification started, days	9 (7–11)	7 (6–8)	<0.001
Parenteral			
Protein, g/kg/day	2.3 (2.1–2.7)	2.0 (1.8–2.3)	<0.001
Carbohydrates, g/kg/day	7.4 (6.1–8.4)	6.2 (5.3–6.8)	<0.001
Fat, g/kg/day	1.5 (1.3–1.8)	1.5 (1.2–1.8)	0.712
Energy, kcal/kg/day	48 (44–56)	43 (38–47)	0.003
Enteral *			
Protein, g/kg/day	3.3 (2.9–3.7)	3.6 (3.1–3.9)	0.148
Carbohydrates, g/kg/day	14.6 (11.5–15.7)	15.1 (13.8–15.8)	0.376
Fat, g/kg/day	5.8 (5.1–7.0)	5.9 (4.9–6.8)	0.469
Energy, kcal/kg/day	129 (112–142)	131 (122–142)	0.714
Total			
Protein, g/kg/day	3.1 (3.0–3.4)	3.2 (3.0–3.4)	0.908
Carbohydrates, g/kg/day	13.4 (11.4–14.1)	12.9 (12.1–13.8)	0.655
Fat, g/kg/day	5.0 (4.6–5.8)	5.0 (4.4–5.7)	0.503
Energy, kcal/kg/day	114 (105–125)	114 (107–120)	0.886
Protein/NPE ratio, g/100 kcal	3.1 (3.0–3.4)	3.1 (2.9–3.4)	0.872

NPE—non-protein energy. * Due to lack of mother’s milk composition data during the first week, enteral intakes from both own mother’s and donor milk are calculated for weeks 2–4.

**Table 3 nutrients-16-01462-t003:** Associations between early nutrient intake and body composition indices estimated by DXA at discharge and sum of SFT at 12 months of CA.

Extremely Preterm, *n* = 32										
	DXA Mass, g		DXA LM, g		DXA FM, g		DXA FM, %		∑ SFT, mm	
	R^2^ = 0.328		R^2^ = 0.319		R^2^ = 0.183		R^2^ = 0.095		R^2^ = 0.192	
	β_s_	*p*	β_s_	*p*	β_s_	*p*	β_s_	*p*	β_s_	*p*
Protein, g/kg/day	−0.04	0.891	−0.11	0.672	0.07	0.818	0.11	0.717	−0.27	0.373
Carbohydrates, g/kg/day	−0.08	0.772	0.05	0.855	−0.20	0.512	−0.22	0.490	0.36	0.287
Fat, g/kg/day	0.32	0.126	0.25	0.246	0.29	0.214	0.21	0.390	0.15	0.559
Sex	−0.30	0.095	−0.42	0.021	−0.03	0.874	0.13	0.534	−0.10	0.629
PMA on DXA day, weeks	0.43	0.018	0.37	0.039	0.39	0.092	0.18	0.367	N/A	
**Very Preterm, *n* = 54**										
	**DXA mass, g**		**DXA LM, g**		**DXA FM, g**		**DXA FM, %**		**∑ SFT, mm**	
	R^2^ = 0.361		R^2^ = 0.312		R^2^ = 0.298		R^2^ = 0.233		R^2^ = 0.163	
	β_s_	*p*	β_s_	*p*	β_s_	*p*	β_s_	*p*	β_s_	*p*
Protein, g/kg/day	0.05	0.795	0.17	0.344	−0.13	0.497	−0.24	0.213	−0.25	0.344
Carbohydrates, g/kg/day	−0.02	0.910	0.02	0.931	−0.06	0.769	−0.04	0.860	−0.09	0.736
Fat, g/kg/day	0.16	0.240	−0.09	0.507	0.39	0.007	0.43	0.005	0.41	0.021
Sex	−0.25	0.061	−0.30	0.030	−0.08	0.549	0.00	0.977	0.09	0.565
PMA on DXA day, weeks	0.43	0.001	0.32	0.021	0.39	0.006	0.27	0.057	N/A	

PMA—postmenstrual age, DXA—dual X-ray absorptiometry, LM—lean mass, FM—fat mass, ∑ SFT—sum of skinfold thickness at 12 months of CA.

## Data Availability

The data generated and analysed during the current study are not publicly available but are available from the corresponding author on reasonable request.

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
