# Peer review of "Impact of Nutrient Intake on Body Composition in Very Low-Birth Weight Infants Following Early Progressive Enteral Feeding"

_nutrients, 2024, doi:10.3390/nu16101462_

Round 1
Reviewer 1 Report
Comments and Suggestions for Authors
I appreciated to review this interesting and novel manuscript. The writing was fluent and sounds scientifically. The authors aimed to prospectively enrolled preterm infants and analyzed their nutrition intake from human milk with a protocol, namely early progressive enteral nutrition, with the body compositions at two time point at discharge (around PMA age 36 weeks) and 12 months corrected age. This manuscript was based on a published article, which previously analyzed the nutritional intake with anthropometric parameter at discharge.
The study designed overall were appropriate. The milk nutrient measurement was standard and the body composition measurement using DXA was novel in preterm infants. The authors divided the study population into two groups extremely preterm infants( EPI) and very preterm infants (VPI) and compared the outcomes between two groups. Generally, the data in table 1 and table 2 were appropriate. The data revealed the key clinical performance of the study unit was good.
In general, I love this article. However, unfortunately, the authors did not control one important factor, postmenstrual age (PMA) at DXA. I understand it was not practical to control the exactly same PMA at DXA due to clinical condition. For this important issue in this novel idea article, the author must explain or use statistical method to control this important factor before publication.
Major concern
Point one:
The figure 2 looks beautiful, however, this figure may give wrong message to readers not in neonatology area. For periviable infants, whose GA were 23 -25 with unstable physical status, they may need larger age to discharge or receive DXA. Hence, for infants with GA 23‑25 weeks, their body weight (in table 3) or estimated body weight were larger (Figure 2a) than infant with GA 30-34 weeks. The fat mass and fat mass percentage may be related to lager postmenstrual age (PMA) at DXA test. Hence, authors MUST show two additional charts before Figure 2a in figures 2. One is PMA received DXA (Y axis) vs GA and the other one is the body weight ad DXA vs GA. Whatever may come, the data must be presented.
Point two:
I suggest to move Table 3 to supplemental data or omit it. As aforementioned and also the data in table 3, all this differences between extremely preterm and very preterm infant comes from different PMA at DXA. Taking the body weight at DXA for example, extremely preterm infants, who usually have extrauterine growth faltering, have higher body weight than very preterm infants in table 3. This phenomenon may be from different PMA at DEX.
Once using corrected age, for example the last item “skinfold thickness at 12 months of corrected age”, the difference may disappear.
Point three:
In the table 4, postnatal or chronological age at DXA must be controlled in the multivariate analysis model. However, since this limit case number, the author can use PMA to replace GA and postnatal age since PMA= GA+ chronological age.
Actually, the author could pool all case in table 4 and make a new variable (EP/VP), and controlled micronutrients, PMA, sex, and EP/VP. The reader can understand EP/VP may have different response to the DXA test.
Minor concern
The title
In the term “early progressive enteral feeding”, I can not find the explanation in “early”. What does “early” mean?
In the abstract, Beta s could be change to coefficients of correlation. DXA should be spelled out fully.
Reviewer 2 Report
Comments and Suggestions for Authors
This marvelous article demonstrates that low birth weight cannot be easily treated to avoid obesity later on, with nutritional fat being apparently more important than protein. The Methods and Statistics are well-versed and the Results are hands-on. I do have some remarks:
- Table 1: The M/F ratio is vastly different between EP and VP. Could this lead to bias in the results? Girls and women have, on average, a lower basal metabolic rate and a lower energy expenditure.
- Figure 2: Is there any information concerning body hydration or total body water? This parameter is also different between M/F.
